# Effect of Sample Transportation on the Proteome of Human Circulating Blood Extracellular Vesicles

**DOI:** 10.3390/ijms23094515

**Published:** 2022-04-19

**Authors:** Anne-Christine Uldry, Anabel Maciel-Dominguez, Maïwenn Jornod, Natasha Buchs, Sophie Braga-Lagache, Justine Brodard, Jovana Jankovic, Nicolas Bonadies, Manfred Heller

**Affiliations:** 1Proteomics and Mass Spectrometry Core Facility, Department for BioMedical Research (DBMR), University of Bern, 3008 Bern, Switzerland; anne-christine.uldry@dbmr.unibe.ch (A.-C.U.); anabel.macield@gmail.com (A.M.-D.); maiwenn.jornod@gmail.com (M.J.); natasha.buchs@dbmr.unibe.ch (N.B.); sophie.lagache@dbmr.unibe.ch (S.B.-L.); 2Department for BioMedical Research, University of Bern, 3008 Bern, Switzerland; nicolas.bonadies@insel.ch; 3Department of Hematology and Central Hematology Laboratory, Inselspital, Bern University Hospital, University of Bern, 3010 Bern, Switzerland; justine.brodard@insel.ch (J.B.); jovana.jankovic@students.unibe.ch (J.J.)

**Keywords:** circulating extracellular vesicles, pneumatical tube system transport, acceleration forces, vibration, label-free proteomics, clinical blood samples

## Abstract

Circulating extracellular vesicles (cEV) are released by many kinds of cells and play an important role in cellular communication, signaling, inflammation modulation, coagulation, and tumor growth. cEV are of growing interest, not only as biomarkers, but also as potential treatment targets. However, very little is known about the effect of transporting biological samples from the clinical ward to the diagnostic laboratory, notably on the protein composition. Pneumatic tube systems (PTS) and human carriers (C) are both routinely used for transport, subjecting the samples to different ranges of mechanical forces. We therefore investigated qualitatively and quantitatively the effect of transport by C and PTS on the human cEV proteome and particle size distribution. We found that samples transported by PTS were subjected to intense, irregular, and multidirectional shocks, while those that were transported by C mostly underwent oscillations at a ground frequency of approximately 4 Hz. PTS resulted in the broadening of nanoparticle size distribution in platelet-free (PFP) but not in platelet-poor plasma (PPP). Cell-type specific cEV-associated protein abundances remained largely unaffected by the transport type. Since residual material of lymphocytes, monocytes, and platelets seemed to dominate cEV proteomes in PPP, it was concluded that PFP should be preferred for any further analyses. Differential expression showed that the impact of the transport method on cEV-associated protein composition was heterogeneous and likely donor-specific. Correlation analysis was nonetheless able to detect that vibration dose, shocks, and imparted energy were associated with different terms depending on the transport, namely in C with cytoskeleton-regulated cell organization activity, and in PTS with a release of extracellular vesicles, mainly from organelle origin, and specifically from mitochondrial structures. Feature selection algorithm identified proteins which, when considered together with the correlated protein-protein interaction network, could be viewed as surrogates of network clusters.

## 1. Introduction

Extracellular vesicles (EV) are spherical particles that are derived from shedding parts of intracellular compartments or plasma membrane through endosomal or ectosomal pathways, respectively [1]. They are enclosed by a lipid bilayer membrane and stabilized by membrane-associated proteins. EV contain a variety of cellular components including metabolites, proteins, and polynucleotides. Essentially all cells produce EV, which act as inter-cellular transport vehicles that convey highly active biological molecules on a variety of biological systems, for instance in the context of cancer [2], plant-microbe interactions [3], inflammation [4], and coagulation [5]. EV that are shed from tissues and cells with access to the circulating blood become cEV and have the potential to accumulate in the peripheral blood (PB). The multiplicity of cell origins and production sites means that the pool of cEV in PB is an extremely heterogeneous mixture containing a variety of biological effectors, which are involved in tissue regeneration or regulation of the tumor microenvironment [1,6]. cEV are, therefore, increasingly recognized as potential biomarkers and targets for treatment in human diseases.

The poorly understood influence of pre-analytical factors as well as the application of a variety of down-stream read-outs, including analytical methods based on fluorescent-activated cell sorting, RNA expression, vesicle size distribution, or coagulation all challenge the establishment of standardized protocols for cEV isolation in a clinical context [7,8,9]. Some of us (Heller, Braga, and Buchs) have previously developed an untargeted approach of label-free proteomics using nanoflow liquid chromatography coupled to tandem mass spectrometry (nLC-MS2) for the semi-quantitative protein profiling of cEV in human PB. By this means, correlations of cEV quantity and their protein content with arteriogenesis in the human heart muscle were identified [10]. Moreover, by characterizing cEV-associated proteins based on gene ontology terms, known cellular location and cell type specificity, we demonstrated that quantitative proteomics enables profiling of the cell origin of cEV, that a single freeze/thaw cycle of blood samples activates coagulation, and the method of freezing of blood samples causes damage to cEV integrity, as indicated by increasing losses of cytosolic proteins between slow freezing at −80 °C and snap-freezing in liquid nitrogen [11].

Many pre-analytical conditions influence the interpretation of blood analyses. These include mainly patient-based factors but also procedural factors, such as the devices that are used, skills of medical staff, lag time, and temperature [12]. Mechanical forces that are associated with PB transport can have a relevant impact on diagnostic tests as well. Most hospitals are fitted with a pneumatic tube system (PTS) which guarantees a fast and efficient delivery of samples from the clinics to the laboratory. However, the pre-analytical impact that is associated with mechanical forces has to be investigated for all potentially susceptible laboratory tests [13] before samples are transported by PTS [14,15,16]. As examples, Kocak and colleagues could not find any statistically significant impact on blood cell counts or erythrocyte sedimentation and standard coagulation tests between samples that were transported by PTS or human carrier (C) [17]. Correspondingly, Phelan et al. did not detect any influence on hemolysis [18]. In contrast, acceleration forces had a relevant impact on platelet aggregometry [19,20], thromboelastometry, and thrombin generation [21], for which reason samples have to be transported by C for these tests. Due to the biological interconnection of platelets with the coagulation system, it is important to investigate whether the transport mode influences the cEV proteome. Only few groups have systematically addressed this question [22,23,24]. They reported an increase in procoagulant activity, along with an increase of annexin-V positive vesicles. However, rather artificial conditions were used, such as extensive stair walking by a carrier or strong agitations on orbital shakers. To close this gap, we set out to investigate the impact of transport by PTS and C on the cEV proteome in a representative clinical context, by untargeted label-free mass spectrometry and recording of the energy levels that were impacted on the blood samples.

## 2. Results

### 2.1. PTS and C Exhibit Substantially Different Transport Metrics

The visual examination of the 3D accelerations that were measured during the transport of the 12 donors’ samples revealed important differences between C and PTS (Appendix A for one representative example). The acceleration signal of all PTS transports exhibited irregular patterns with several peaks of large amplitude and short duration, which occurred in any possible direction. In contrast, the C samples showed more gentle patterns, regular but complex oscillatory signatures that were typical of a walker (Appendix A). There were three metrics integrating the acceleration signal over time that were calculated for each transport event (see Section 4.2 and Methods section of Appendix A): mean Teaker–Kaiser operator (TK), root mean square (RMS), and vibration dose value (VDV). A boxplot of those transport metrics is shown in Figure 1. Transportation through PTS lasted on average less than half as long as by C (2.4 ± 0.3 min compared to 5.5 ± 0.5 min). PTS subjected probes to substantially higher accelerations than C, with maximum amplitudes per journey > 17 g in PTS and <2.4 g in C. Moreover, PTS signals exhibited a significantly skewed distribution towards higher g-forces with a mean of 116 shocks having an amplitude >2.5 g. Transport metrics TK, RMS, VDV were significantly higher by approximately one order of magnitude in PTS compared to C (Appendix A). In spite of this, the median accelerations of both modes of transport through the transport event time were in a similar low range (0.1–0.4 g). Since the same carrier transported all the samples, it was not too surprising that the ground frequencies in C were all very similar at 3.98 ± 0.07 Hz. This ground frequency appears to be characteristic of the gait and speed of the walker, as tests that were performed with two other carriers gave 2 Hz and 4.3 Hz, respectively (not shown). In summary, transport metrics were substantially different between PTS and C, with low intensity, regular oscillations for C compared to very high, irregular, and multidirectional accelerations of short duration for PTS, respectively.

### 2.2. PFP Centrifugation Protocol Isolates Pure cEV

A total of two plasma preparation procedures were considered in this work (see Materials and Methods), namely platelet-poor (PPP) and platelet-free plasma (PFP). While our laboratory has shown earlier that the applied centrifugation protocol does successfully isolate cEV from PFP [11], the concept behind the use of PPP is that it contains cell fragments, an indicator of differential cell damage in case one transportation mode is more damaging to cells than the other. The demonstration by Braga-Lagache et al. [11] that the centrifugation protocol applied to PFP does indeed isolate cEV was based on mathematical vesicle sedimentation modelling and transmission electron microscopy imaging; it was shown that larger vesicles of diameters >500 nm are almost entirely removed by the short high-speed centrifugation of PPP, and part of the smallest vesicles (<200 nm) are lost due to their low sedimentation speed. Here, the same operator applied the exact same cEV isolation procedure using the same centrifuges as in Braga-Lagache et al. In order to confirm that the conclusions that were drawn earlier [11] still apply here, we reprocessed the 400 μL PFP data, consisting of 12 healthy donors, with the same data interpretation pipeline as described here in the Materials and Methods section and made a correlation analysis of the log2-transformed median protein intensities of proteins that were quantified at least three times in both datasets (N = 1009, Appendix A). The squared correlation coefficient (R^2^) of 0.49 of the cell surface protein class was poor, but the serum/plasma, cell part, and cell membrane proteins scored with R^2^ of 0.70, 0.77, and 0.75, respectively. The cell surface protein class contained the least members (N = 70) and had three extreme outliers in form of isoform 2 of ficolin-3, hornerin, and platelet factor 4 variant-1. By excluding these three proteins, the R^2^ value increased to 0.75.

Due to the large number of overlapping proteins showing a good correlation of their intensities with Braga-Lagache et al. data [11], we can conclude that the particles that were isolated from PFP in this study are true and pure cEV.

### 2.3. Nanoparticle Size Distribution Is Influenced by Transport in PFP

The particle size distribution of all samples was measured in order to characterize the potential differences that were conferred by plasma preparation and/or transport method. Visual inspection revealed that most of the donor samples had a particle size distribution peaking in the range of 70–130 nm (Appendix A). We note that this is somewhat lower than the 200 nm that was determined in an earlier study [11] using cryo-transmission microscopy imaging of isolated cEV. We also note in Appendix A that in PFP increased irregularities were seen in PTS compared to C in PFP. Figure 2 presents an overview of features that were extracted from the ZetaView^®^ distributions. Inspection of the AUC per donor showed that we identified consistently less particles in the PPP samples compared to the PFP, with the exception of donor BE351, who generated the two outliers in the PPP AUC plot. Generally, we found more differences between the transportation methods using PFP (upper row of Figure 2) compared to PPP (lower row of Figure 2). In PFP the particle size distribution in PTS compared to C was (figures in brackets are mean ± standard deviation) (i) wider (113 ± 14 vs. 66 ± 10 nm), (ii) less skewed (1.7 ± 0.4 vs. 2.7 ± 0.9 nm), and (iii) peaking at a larger size (116 ± 21 vs. 84 ± 7 nm). In PFP, PTS also had a larger median particle size distribution than C (134 ± 13 vs. 100 ± 5 nm). Again, healthy donor BE351 was an exception with similar medians in both transportation methods. The particle numbers were not significantly different (8 × 10^12^ ± 2 × 10^12^ vs. 6 × 10^12^ ± 3 × 10^12^), although we observed in PFP a trend towards more particles in PTS compared to C. No significant differences of the ZetaView^®^ metrics were identifiable in the PPP samples.

In summary, using a nano-particle detection method, we detected counter-intuitively higher numbers of particles in PFP compared to PPP. We also saw a wider size distribution in PTS compared to C, but this transport effect on the cEV particle distribution is only detectable in PFP. We concluded that non-cEV plasma constituents are a pre-analytical confounding factor for nano-particle detection technology.

### 2.4. cEV Isolated from PPP Are Contaminated by Platelet, Lymphocyte and Monocyte Remnants

A total of 2216 protein groups were identified by mass spectrometry when combining all PFP and PPP samples together; 2144 of them were detected in at least two out of the three technical replicates of at least one donor and were further considered for analysis. In general, the number of quantified protein groups were lower in PFP compared to PPP (Appendix A). The only exception was donor BE140, for whom we could quantify 209 more proteins in PFP. An overview of the number of protein groups that were found in each protein class and category is shown in Table 1a, while Table 1b gives the numbers per cell type; the complete annotated protein list is in Appendix A. We observed that the total number of cell type specific markers is very similar in PFP and in PPP, and so is the number of serum/plasma proteins. However, PPP had a markedly higher number of cellular class proteins (membrane, cell part, cell surface) than PFP. In total, there were 52 protein groups that were exclusively quantified in PFP (unique-to-PFP) and 456 in PPP (unique-to-PPP). We found 12 platelet-specific proteins in the unique-to-PFP and 393 in the unique-to-PPP set, resulting in a much higher ratio for platelet-specific proteins in PPP (393/456 = 86.2%) compared to PFP (12/52 = 23.1%). A further confirmation of the prevalence of platelet remnants in PPP is that four out of the six unique-to-PPP markers were of platelet origin (*CD93*, *CD224*, *CD244*, and *CD274*) [25], while only one out of the six unique-to-PFP markers was with a platelet annotation (*CD81*, as listed in Appendix A).

Additional information regarding the cell origin of differentially abundant cEV was gained by comparing PFP_C, PFP_PTS, PPP_C, and PPP_PTS iTop3 intensities of a choice of proteins that can be regarded as cell-type specific (Table 2 and Figure 3). The top row of Figure 3 shows the markers that are enriched in PFP compared to PPP, the bottom row those that are enriched in PPP compared to PFP; we note that the enrichment in each case is significant for both transport modes. The top row consists of specific markers for erythrocytes (*CD233*), macrophages (*CD14*), endothelial cells (*HSPG2*), and exosomes (*CD81*). The markers in the second row can be catalogued as platelets (*CD41*, *CD62P*), as well as lymphocytes and monocytes (*CD40*, *CD102*). We noted no significant intensity difference between PTS and C, neither in PFP nor in PPP; this subject is treated in the next section. Although this was not confirmed by nanoparticle tracking, one can assume that, independently of the mode of transport, larger vesicles that are derived from cell damage were present in PPP, but had been removed in PFP by the second centrifugation step. We can therefore interpret any transport-independent increase in PPP as stemming from cell fragments, while an increase in PFP can be seen as originating from actual cEV.

In summary, we can conclude that (i) cEV that are isolated from PPP contain more platelet remnants and lymphocyte/monocyte components compared to PFP, independently of the transport method, and (ii) the increased intensity in PFP of erythrocyte (*CD235a*), endothelial cell (*HSPG2*), macrophage (*CD14*), and the exosomal marker *CD81* indicates an enrichment of true cEV proteins in PFP. Overall, our data let us conclude that further analyses should be focused on the purer PFP-derived cEV.

### 2.5. Non-Consistent Impact of Transport Method on Individual cEV Protein Compositions

We had hypothesized that the differential cell damage that is caused by the modes of transports would be detected in the cEV proteome and be interpretable as different formations of cell debris or stimulations of blood cells (especially platelets). However, the differential protein quantification analysis of PFP samples showed, in all donors except BE351, only very few significant differences between PTS and C (Appendix A and summarized in Appendix A). There was just one, albeit different, protein that was enriched in C of BE354 (healthy) and BE363 (secondary AML), and five that were enriched in PTS of BE354; the case of the outlier BE351 is discussed in Supplementary Results. We then looked in more detail at the behavior of the detected CD markers; their relative PTS to C changes (log2 fold change) is shown, per donor, in Appendix A. The plots revealed that PTS transport enriched for erythrocyte-derived cEV (*CD233*) in half of the donor samples, independently of the plasma preparation. No other trend seemed to emerge from any other markers. Interestingly however, the platelet markers *CD41* and *CD62P* appeared well correlated: if one marker was enriched, respectively depleted, for one donor, the other marker was enriched, respectively depleted as well.

While very few changes in protein abundance turned out to be significant, there were nonetheless a number of proteins that were not detected in either C (median of 21, range 6–271) or PTS (36.5, 8–584) (Appendix A). A vast majority of those on-off proteins were detected only once (92% in PTS with a total of 487 on-off proteins, and 73% in C with 575 proteins), and 7% and 24% detected twice, respectively (excluding BE351). The fact that those on-off proteins were randomly occurring between the donors and did not reach statistical significance indicates low intensities, therefore can be considered as being missed by chance during mass spectrometric analysis.

As BE351 and BE140 were identified as outliers by particle size distributions and enrichment of proteins in PFP or PTS, we argue (Supplementary Results) that cEV damages in these two cases occurred during blood collection or processing of the sample. For this reason, we decided to remove these samples from the subsequent analysis.

In summary, no consistent statistical protein differential expression could be detected in PFP between PTS and C.

### 2.6. Blood Cell Counts and Nanoparticle Features Do Not Correlate with cEV-Associated Protein Intensities

A question of interest is whether cEV-associated protein intensities, aggregated by relevant subclasses or annotations, are able to predict the number of nanoparticles, hemoglobin concentrations, or blood cell counts as determined by Sysmex and ZetaView^®^. To this purpose, Spearman’s rank correlations were calculated between all these quantities, including transport metrics as well, based on the 10 remaining donor values and focusing exclusively on PFP (Appendix A). The result, in the form of an unsupervised cluster of correlation coefficients, is shown in Figure 4. Anti-correlations are colored blue, positive correlations range from green to red as the coefficient increases. There are four distinct clusters of various sizes that are discernible in this picture. The largest cluster (top right corner) was formed by cell-derived protein classes and cell markers. Leukocyte and granulocyte cell markers correlated weakly, respectively not at all, with cell part, cell membrane, cell surface, and exosome protein abundances. Platelet markers on the other side showed good to very good correlations with these features, indicating that a large part of cEV were probably of platelet origin. Interestingly, coagulation factors were part of this cluster too; they correlated with platelet markers, suggesting that such factors are indeed associated with platelet-derived cEV. We additionally noted that the cEV-associated protein intensities did not correlate with blood cell count or nanoparticle features.

The second largest cluster (lower left to center) was formed by blood cell counts, including hemoglobin (*HBG*), and surprisingly by apolipoproteins intensities that were determined by our proteomics approach. Apolipoproteins correlated with all cell type counts except granulocytes (Gc) and monocytes (Mc). Erythrocyte (Ec) cell counts correlated weakly with the erythrocyte cell markers that were quantified by proteomics.

The small 2 × 2 cluster on the lower left corner consisted of proteins that were annotated with the GO term “blood microparticles” and immunoglobulins. The small size of this cluster comes as a surprise, as one would expect more correlations with “blood microparticles” in this context.

The last cluster (middle of lower left quadrant) was formed by a sub-cluster of the transport metrics and a sub-cluster of values that were derived from the nano-particle tracking system. Interestingly, the two sub-clusters were fused via the calculated particle volume, which significantly correlated with the transport metrics VDV, RMS, and TK, while the particle concentration also correlated with the VDV metric. This corroborates somehow our observation that PTS transport does lead to a widening of particle size, hence transport mode can have an influence on particle volume in PFP as stated above (Figure 2, Appendix A).

Furthermore, particle concentrations showed a correlation with “other plasma proteins”, which included serum albumin and alpha-2-macroglobulin, as two examples with high proteomics determined intensity and of larger molecular weight. Additionally, the original particle volume correlated with the proteomics-derived apolipoprotein intensities and lymphocyte cell counts. While the latter is difficult to explain, the apolipoprotein and other plasma protein correlations do indicate that they may play a role as a confounding factor in nanoparticle tracking measurements as already indicated in above Section 2.3. We also noted that keratin did not correlate with any other features, which supports the notion that keratins may be contaminants rather than products that are released into cEV.

In summary, we concluded that a significant proportion of cellular proteins in the cEV fraction of PFP is originating from all blood cell types, but with the exception of erythrocytes there is no general correlation between cell counts in blood and the corresponding proteomics-based cell enumeration in cEV.

### 2.7. Transport Metric Correlations and Lasso Reveal Specific Effects on cEV Proteome

While no statistically significant groups of proteins emerged from the differential analysis between PTS and C, consistent abundance changes correlating with transport metrics can provide insight into the impact of transport on the cEV proteome. Spearman’s rank correlation tests were, therefore, performed between the transport metrics (TK, RMS, and VDV) and all the detected protein intensities. The number of protein groups correlating, or anti-correlating significantly with either one, two or three transport metrics is reported in Table 3. C and PTS results were at first pooled together (C+PTS), then considered separately. We noticed indeed that by considering C and PTS together we had three to five times less correlations than by looking at C or PTS individually, an indication that C and PTS followed distinct patterns. Another observation was that in C a high number of proteins correlated concurrently with all the transport metrics, while in PTS two distinct groups of proteins were seen, one correlating only with VDV and another one with both TK and RMS. The same applied to the negatively correlating proteins, albeit with much fewer proteins involved. We established three ranked lists for GO term enrichment analyses (see the Methods section), containing the proteins correlating (i) in C with all three metrics TK, RMS, and VDV (C_TK/RMS/VDV); (ii) in PTS with both TK and RMS (PTS_TK/RMS); and (iii) in PTS with VDV alone (PTS_VDV). The resulting unique GO term list with the calculated *p*-values and the numbers of proteins/gene products is given in Table 4. Based on these GO terms, it appeared that increased oscillations associated with C induced an increased level of cEV-associated proteins which regulated the cytoskeleton and were involved in cellular organization (including cell projections and anchoring junctions). An increase of shocks, as recorded by TK and RMS during PTS transport, appeared on the other hand to induce an increased level of cEV from organelle origin, and of cEV that is involved in the metabolism of proteins and nucleic acids. High vibration doses (VDV) during PTS transport seemed to have a specific impact on mitochondrion-associated cellular respiration, together with increased membrane assembly by cell junction proteins.

The least absolute shrinkage and selection operator (Lasso) algorithm was applied in order to select EV-associated proteins that could classify the transport type. Although the leave-one-out misclassification error was high (Appendix A), the global response (predictor) provided a good separation between C and PTS, in particular with the pure Lasso approach. Between the pure Lasso and the elastic net approaches, we identified altogether 12 proteins that could be used as classifiers (Appendix A); five had a positive (*CFHR1*, *KRT1*, *FLOT1*, *HRG*, *SERPINC1*) and seven a negative (*CORO1A*, *ATP5PF*, *ST6GAL1*, *HSPA1A*, *EFEMP1*, *OIT3*, *C4BPB*) coefficient. Complement factor H-related protein 1 (*CFHR1*), histidine-rich glycoprotein (*HRG*), and antithrombin-III (*SERPINC1*) are proteins that are adherent to the extracellular matrix and secreted. *SERPINC1* is localized in the endoplasmatic lumen and a inhibitor of coagulation, HRG acts as a versatile adaptor protein regulating, amongst others, cell adhesion processes, and *CFHR1* is involved in the complement regulation. *KRT1* (keratin type II cytoskeletal 1) and *FLOT1* (flottilin-1) are associated with cellular membranes, with *FLOT1* cooperating in the process of caveolae-like vesicle formation, while *KRT1* may regulate the activity of kinases via binding to integrin beta-1. Among the proteins with a negative coefficient, Coronin-1A (*CORO1A*) is part of the cytoskeleton that is involved in the invagination or protrusion of plasma membrane, and mitochondrial ATP synthase-coupling factor 6 isoform 2 (*ATP5J* or *ATP5PF*) is part of the ATP-synthase complex at the inner membrane of mitochondria, respectively. Oncoprotein-induced transcript 3 (*OIT3*), C4b-binding protein beta isoform 2 (*C4BPB*), beta-galactosidase alpha-2,6-sialyltranferase 1 (*ST6GAL1*), and EGF-containing fibulin-like extracellular matrix protein isoform 2 (*EFEMP1*) are annotated as being secreted, but are also found with the following cellular association: the extracellular matrix for *C4BPB* and *EFEMP1*, the Golgi apparatus for *ST6GAL1*, and the nucleus envelope for *OIT3*, respectively.

The elastic net method retained only three proteins, two of which, *ST6GAL1* and *SERPINC1*, had already been selected by the pure Lasso procedure. The remaining one, *HSPA1A* (heat shock 70 kDa protein 1A), is a multi-functional protein located at many sites within a cell, namely the cytoplasm, cytoskeleton, microtubule organizing center, and centrosome. *CORO1A* and *FLOT1* were also proteins significantly correlating in C with all transport metrics (C_TK/RMS/VDV) or in PTS with the vibrational dose values (PTS_VDV).

In summary, the Lasso proteins directed towards an association of cellular organelles with transport metrics, supporting the correlation analysis between protein intensity and the transport metrics as shown above.

### 2.8. Protein-Protein Interaction Network Conclusions

A STRING network analysis was then performed in order to study the interactions between the proteins of interest that were identified by the correlation analysis and the Lasso algorithm. The Lasso proteins were added to each of the three correlation lists C_TK/RMS/VDV, PTS_TK/RMS, and PTS_VDV and the three completed lists were submitted to the STRING database. A first impression that was given by all three networks was that the Lasso proteins were distributed throughout the network in different clusters. In order to highlight this fact, the three STRING networks, which were based on combined confidence score levels ≥ 0.70, are visualized in Figure 5, Figure 6 and Figure 7 after the application of the GLay community clustering algorithm [26]. When separated in clusters by this algorithm, we noted that apart from *SERPINC1* and *HRG*, who are known interactors, the Lasso proteins were indeed spread out through different community structures of the networks. The Lasso proteins were identified in the figures by a red border, and all the proteins were annotated with a selected minimal GO term list, as explained in the Methods section (color coded as given in Table 4). The clusters often had a dominant set of GO terms/colors, and most clusters integrated at most one, in some cases up to three Lasso proteins. Proteins with no interaction partners were omitted; this was the case for the Lasso proteins *OIT3*, *C4BPB*, *ST6GAL1*, and *EFEMP1*. *CORO1A*, *CFHR1*, and *KRT1* only have interactions in the PTS_VDV network, while *SERPINC1*-*HRG*, *ATP5PF* (aka *ATP5J*), *HSPA1A*, and *FLOT1* were part of all three networks.

In the C_TK/RMS/VDV network, *SERPINC1*-*HRG* were attached to a cluster dominated by anchoring junction proteins (color 8); *ATP5J* was part of a small cluster of regulation of cellular component proteins (color 1); *HSPA1A* sat in a cluster with many establishment of localization in cell (color 3) and nitrogen compound transport proteins (color 4); and *FLOT1* and *CORO1A* were both linked independently to a multi-component cluster of regulation of cellular component organization (color 1), cytoskeleton organization (color 2), actin filament-based process (color 6) and actin cytoskeleton proteins (color 9).

In the PTS_TK/RMS network, *SERPINC1*-*HRG* were not attached to any further proteins; *ATP5J* formed a hub between organelle envelope proteins (color 4); *HSPA1A* was integrated in a cluster that was dominated by organelle organization (color 1); *FLOT1* was connected to a single protein annotated to the latter term.

In the PTS_VDV network, *SERPINC1*-*HRG* as well as *FLOT1* were attached to a cluster that was dominated by cell junction proteins (color 6), although the connection by SERPINC1-HRG is made through a non-distinct subunit of intrinsic components of membrane proteins (color 5); *CFHR1* was also linked to a predominantly cell junction cluster of which *CORO1A* was also part of (color 6), and *KRT1* to a predominantly intrinsic component of membrane cluster (color 5); *ATP5J* was associated to clusters of many mitochondrial protein-containing complex (color 8) and cellular respiration proteins (color 3); *HSPA1A* was included in a cluster that was dominated by cellular localization proteins (color 1).

In summary, the Lasso algorithm identified a set of proteins that can be regarded as representing the community clusters that were extracted from the protein-protein interaction networks created from the transport metric correlation analyses.

## 3. Discussion

To the best of our knowledge, only a few studies have investigated the influence of transport on cEV integrity and composition, with important limitations in the applied experimental design. Lacroix et al. [22] attempted to measure the impact of carrier transportation. They studied five modes of transportation: (i) gentle tube conversion, (ii) strong agitation by rotating the tubes for two hours on a wheel, and human carrier transport three floors down in tubes, (iii) unsupported, (iv) horizontally, or (v) vertically fixed in a box. All the tubes were incubated for two hours at room temperature before centrifugation, which is not standard clinical practice and might have introduced a bias in the study results. They measured an increase in annexin-V positive microvesicles (analyzed by flow cytometry) and an increased procoagulant activity in the case of strong agitation and carrier transport, except when the tubes were kept fixed in a vertical position. Gyorgy et al. simulated transport by 50 Hz amplitude on an orbital shaker for one hour at 37 °C [23] and Baek et al. used 450 rpm for one hour at RT [24]. Gyorgy et al. also found increased annexin-V-positive microvesicles by flow cytometry and a significant increase of vesicles by agitation in citrated blood, similar to Lacroix et al. Baek and colleagues used a protein microarray-based analysis platform, which detects exosomes based on binding to several cluster of differentiation markers and annexin-V. They determined a not statistically significant tendency of increased exosome binding with citrated blood after agitation. Overall, all these studies had a very artificial design with long blood incubation times and application of forces and oscillation frequencies that are not occurring during routine clinical transport scenarios.

A more comprehensive and clinically appropriate study is presented here. Blood samples from twelve donors, six of which with hematological disorders, were transported from the identical collection point to the wet lab both by either a foot carrier (C) or by the pneumatic tube system (PTS) of the hospital. The transport forces were measured and recorded. Both PPP and PFP plasma types were prepared so as to investigate the impact of transportation on cEV on both plasma preparation methods. Nanoparticles were analyzed by ZetaView^®^ and the protein profile was determined with a semi-quantitative, label-free proteome approach. We discuss in the following some of the caveats of the methods that were considered and the conclusions drawn from the study.

The isolation of cEV can be a difficult task, as blood contains other sorts of nanoparticles or large protein complexes which are difficult to separate from the cEV by physical means. For this aim, we chose an earlier established centrifugation protocol [11], where we have shown that several consecutive washing steps in PBS are suitable for cEV isolation and subsequent proteome analysis; the procedure, however, does not eliminate some major plasma protein contaminations such as lipoproteins, coagulation, or complement factors as well as immunoglobulins. Size exclusion chromatography (SEC) has been recommended as an alternative isolation option with good recoveries of EV [7]. We have tested SEC and compared the resulting cEV proteomes with the ones that were achieved with our centrifugation-based method (see Supplementary Methods and Results). We found that SEC did not result in more specific cEV isolation. Furthermore, the reproducibility was also compromised, when compared with our centrifugation protocol (Appendix A). For these reasons we used our earlier developed protocol for this study, although one might ask whether it makes any sense to analyze differences that are caused by transport accelerations, when the samples are later subjected to much higher accelerations in the centrifuge. We must remember, however, that centrifugation generates a mostly constant and unidirectional acceleration; this is qualitatively completely different to transportation, where the samples are shaken and hit from any possible direction. Perhaps more importantly, while it can be possible that centrifugation does alter the size distribution and eventually the cEV composition, all the samples experienced the same treatment, hence the impact on cEV integrity during the centrifugations was the same for all the samples.

As suspected, the measurement of mechanical forces during transport revealed extremely different patterns depending on the transport mode: C samples were subjected to regular oscillations of moderate amplitude, while PTS samples were subjected to successive shocks of high amplitude. The relatively rough transport of blood specimens through the PTS has been known to affect thrombin activation [21]. Otherwise, only little is known about the molecular impact on other blood components. The gentler transportation by a human courier appears as the more adequate way of transporting blood from bedside into laboratories for subsequent molecular characterization, which was advocated in the past by different laboratories that were interested in the analysis of cEV. One interesting point, however, should be noted: in our study we observed a higher relative variability of the mechanical forces in C compared to PTS (relative standard deviation RSD in Appendix A), meaning that the impact on cEV-associated proteins is relatively more variable in C than in PTS. PTS would, therefore, be the preferable choice in the context of using the cEV proteome composition in biomarker discovery projects. The RSD that was obtained here was of course conditioned by the design of this study, with blood samples all sent along the same path each time.

Based on our nanoparticle tracking results, we concluded that strong transport forces occurring during PTS induced the spreading of the cEV size distribution towards both smaller and larger particles, with a shift of the median size to larger values (Figure 2). This observation can be explained by the destruction of some vesicles, which end up forming smaller fragments, together with the concomitant formation of larger aggregates due to the possible activation of thrombin [21]. The significant correlation between all three calculated transport metrics with the particle volume that was derived from nanoparticle tracking measurements underlines that the formation of particle aggregates might be the biological phenomenon that could explain this finding (Figure 4). Other observations were that (i) PTS-induced particle spreading in PFP could not be detected in PPP, (ii) the area under the curve increased in PFP compared with PPP (Figure 2, Appendix A), and (iii) there exists a correlation between the original particle concentration that was determined by ZetaView^®^ and apolipoproteins and other plasma proteins (Figure 4). The increase of the measured number of particles when the plasma contaminations are actually partially removed by additional centrifugation indicates that confounding factor(s) in plasma suppress(es) the detectability of nanoparticles (see also Appendix A). Particles that were removed by additional centrifugation are most likely lipoproteins and aggregates of plasma proteins.

We showed with our label-free proteomics approach that the different modes of transportation do not have an impact on the plasma membrane-embedded cell type-specific cEV-associated proteins, but rather on proteins from subcellular structures, such as organelles, their membranes, and the cytoskeleton. However, the individual patterns were highly heterogeneous and distinct between different donors. The GO term analysis revealed an increase in the cytoskeleton-regulated plasma membrane and cell organization activity that was caused by gentle oscillatory forces during C, in contrast to the release of intracellular vesicles from organelles, including mitochondrial structures, that was caused by the high energy vibration dose during PTS transports.

It is generally assumed that mechanical forces induce activation of platelets. One could, therefore, expect that the number and intensity of platelet-derived proteins correlate with the recorded energy impacting on blood during PTS transportation. However, we could not confirm such a trend (Figure 3 and Figure 4, Appendix A). Moreover, we did not measure a significant correlation between the cellular protein intensities and the platelet numbers, neither in cEV that were isolated from PFP (Figure 4), and more intriguingly not in PPP (not shown), where more platelet fragments are present (Figure 3). In fact, we could only find a significant correlation between the proteome-based quantification of erythrocyte origin, based on erythrocyte-specific cell surface proteins, and erythrocyte numbers that were measured in whole blood, which might indicate that cell type-specific proteins from cEV do not reflect the concentration of cell types in blood, with the exception of erythrocytes (Figure 4).

On the other hand, existing cEV might be destroyed by higher energy that is associated with harsher transport conditions, resulting in a decrease of cellular proteins in the cEV fraction of blood. With the cell origin profiling, we found no generalizable pattern between the blood samples from the twelve donors. We, therefore, conclude that transport-related forces do not have a consistent impact on cEV composition, and that cEV integrity is an individual trait. Additionally, there might be other underlying factors that we do not yet fully understand, as for instance the aggregation of cEV that is induced by freeze-thawing of PFP and during isolation by centrifugation, as observed earlier [11].

## 4. Materials and Methods

### 4.1. Study Design, Study Participants, Blood Sampling and Ethics Approval

This is a monocentric, exploratory study using peripheral blood (PB) that was transported either by the hospital pneumatic tubing system (PTS) or a human courier (C). Platelet-poor (PPP) or platelet-free (PFP) plasmas were subsequently prepared and the nanoparticle size distribution and protein composition of the isolated cEV were analyzed for each of the four combinations of plasma type and transport mode PPP_C, PPP_PTS, PFP_C, and PFP_PTS.

A total of six hematologically healthy volunteers (three females and three males between age 39 and 56, mean of 46) and six patients with myeloid malignancies (one female and five males between age 31 and 82, mean of 60) were included in this study (Appendix A). The PB of each donor was drawn by venipuncture at the same ward in the hospital and collected into four 4.3 mL S-Monovette 3.2% citrated tubes, plus one 4.3 mL S-Monovette EDTA tube (Sarstedt, Germany). Of these, two of the citrated S-Monovettes were immediately sent to the proteomics laboratory through PTS, including a sensor unit that was fixed within the transport tube. The same sensor unit was subsequently attached to a recipient rack holding the two remaining citrated and the EDTA S-Monovettes in an upright position. The same carrier walked the samples with the attached sensors to the proteomics laboratory following the same route as much as possible. Immediately after transportation, the EDTA sample was used for blood cell counting with a Sysmex XN-1000 instrument (Sysmex Suisse AG, Horgen, Switzerland). Clinical data were collected on the Swiss Myelodysplastic Syndromes (MDS) Registry/Biobank platform, where patients with MDS, acute myeloid leukemia (AML), and healthy volunteers were included. MDS patients were risk-stratified according to IPSS-R using a cut-off between 4 and 4.5 points for lower and higher risk disease [27].

### 4.2. Determination of Transport Metrics

A sensor unit consisting of a Raspberry Pi Zero mini-computer that was equipped with a SenseHat (RPi0-SH) add-on board was used for the acceleration measurements. The acceleration events over time during transport of a sample can be summarized by different metrics. There are three transport metrics that are commonly used in the context of shocks and vibrations that were extracted (detailed in the Methods section of Appendix A): (i) mean Teaker–Kaiser operator (TK), a measure of the mean energy of the signal; (ii) root mean square (RMS), a measure of the mean acceleration; and (iii) vibration dose value (VDV), a quantifier for the sum of vibration events. Further transport features were determined, such as duration, ranges, and distribution of signals. Since the C transports exhibited several periods of regular, sustained oscillations, a period of 0.5–3 min duration was chosen for each C transport and the corresponding signal Fourier-transformed in order to extract the ground frequency. The calculations were performed using *base R* functions (version 3.6.3) as well as *caTools*, *e1071* and *signal* packages. There are two sensor limitations that have to be noted. Firstly, the sampling rate of the sensors is such that no frequency higher than 12 Hz can be measured. Tests with a different accelerator of higher sampling rate did not, however, detect substantial Fourier components in the 15–32 Hz range (not shown). Secondly, the peak values that were measured during PTS were at the upper limit of the detection power of the accelerometer, and therefore, higher accelerations may have occurred.

### 4.3. Reagents, Software and Data

All the reagents were of analytical purity grade. Dithiothreitol (DTT), iodoacetamide (IAA), and LC-MS grade acetonitrile were purchased from Fluka (Buchs, Switzerland); urea, trifluoroacetic acid (TFA), and formic acid from Merck (Zug, Switzerland); TRIS and acetone from Sigma (Buchs, Switzerland); and sequencing-grade endoproteinase LysC and porcine trypsin from Promega (Dübendorf, Switzerland). Phosphate-buffered saline solution was from Gibco (Life Technologies, Zug, Switzerland) and sterile filtered through 0.2 μm pore size membrane (Millipore, Zug, Switzerland).

Normalization, imputation, statistical tests, and Spearman rank correlations were calculated using *base R* with following additional packages: *vsn*, *MSnbase,* and *limma*. All protein expression data are provided in the Appendix A. *Lasso* feature selection was performed using the R package *glmnet* (version 4.0-2). Community clustering was calculated by the GLay app [26] from Cytoscape [28]. Graphics art were designed in Photoshop using figures that were produced with *R,* Excel, and Cytoscape.

### 4.4. Preparation of Platelet-Poor (PPP) and Platelet-Free Plasmas (PFP)

We prepared plasma samples as PPP and PFP from both citrated S-Monovettes that were transported either by C or PTS. All the samples were centrifuged in a swing out rotor (Labofuge 400R function line) at 1500 g for 10 min at room temperature to separate the plasma from the cell fraction. PPP was carefully extracted, without disturbing the cellular fraction, leaving 0.5 cm of liquid above the buffy coat. The collected PPPs from the two S-Monovettes deriving from the same transport type were mixed, aliquots of 400 μL were taken, and the remaining volume was distributed in 2mL tubes with a volume of 1.8mL per tube and further centrifuged for 2 min at 16,000g (Eppendorf, centrifuge 5415 R). After this second centrifugation step, the PFP was carefully removed, leaving 50–100 μL back in the tube, mixed, and dispersed into 400 μL aliquots. The PPP and PFP aliquots were frozen at −80 °C until further use.

### 4.5. Nanoparticle Tracking Analysis

Nanoparticle tracking was performed on a ZetaView^®^ (Particle Metrix, Inning am Ammersee, Germany) instrument using an embedded laser (488 nm) and a CMOS camera with the following settings: autofocus on, camera sensitivity at 85, shutter 100, scattering intensity 4, and cell temperature at 25 °C. PPP and PFP were diluted in sterile filtered PBS using a 0.22 μm membrane. Several dilutions were made targeting at least 1000 particle traces. The tracing videos were analyzed by ZetaView^®^ software (version 8.05.05 SP2) limiting the particle size range to 10–1000 nm with a minimum particle brightness of 30. Particle counts and concentrations were averaged from several dilution measurements and the original particle volume was calculated by the particle volume that was corrected by the dilution factor used to dilute the plasma samples before nanoparticle tracking. All distribution features (such as mean particle size, distribution width etc.) were calculated using *base R* functions and the *caTools* package on the binned particle concentration curve. The total amount of particles was defined as the area under the curve (AUC) by applying the trapezoidal integration rule.

### 4.6. Isolation of cEV and Protein Digestion

Isolation of cEV and protein digestion were performed as previously reported [11]. Briefly, PFP or PPP aliquots of 400 μL were slowly defrosted on ice, then centrifuged for 40 min at 16,000× *g* and 20 °C followed by three washing cycles of the resulting pellets with 250 uL PBS and centrifugation for 20 min at 16,000× *g* and 20 °C. The final pellets containing cEVs were dissolved in 10 μL 8 M urea/100 mM Tris*HCl pH 8.0, reduced with DTT, alkylated with IAA, and double digested by a combination of LysC and trypsin protease (100 ng each). From each donor, both transport types, as well as PFP and PPP samples, we isolated cEVs from three different aliquots (technical replicates) resulting in a total of 144 cEV samples for mass spectrometry analysis.

### 4.7. Mass Spectrometry and Label-Free Protein Profiling

Shotgun nLC-MS2 was used in a data-dependent acquisition (DDA) mode. Peptide sequencing was performed on an Orbitrap Fusion LUMOS mass spectrometer that was coupled with a Dionex Ultimate 3000 nano-UPLC system (ThermoFischer Scientific, Reinach, Switzerland) as described elsewhere [29]. Each protein digest was run two times by loading 5 μL onto the pre-column. The mass spectrometry data of all runs (288 files) were processed with MaxQuant/Andromeda (version 1.6.6.0) searching against the concatenated forward and reversed SwissProt human protein database (release 2019_07) with the following parameters: Mass error tolerance for parent ions of 10 ppm and fragment ions of 0.4 Da, strict trypsin cleavage mode with 3 missed cleavages allowed, static carbamidomethylation on Cys, variable oxidation on Met and acetylation of protein N-termini. The match-between-runs option in MaxQuant was allowed only within PFP, respectively PPP samples, by allocating non-consecutive fraction numbers for PFP and PPP. Otherwise, the default MaxQuant settings were used. Identification results were filtered on the peptide spectrum match, peptide, and protein identification level to a 1% false discovery rate (FDR). In addition, only the protein groups that were identified with at least two distinct peptides were accepted. All mass spectrometry data are available via ProteomeXchange (identifier PXD033117).

### 4.8. Protein Classification

The identified proteins were manually classified using information that was retrieved from the Uniprot database in the following manner: transmembrane or intramembrane annotations in conjunction with subcellular location were considered first; if not conclusive or not present, GO:CC terms were considered next, then GO:BP terms; if still no determining terms could be extracted, then either tissue specificity, keywords, or protein description were used instead.

All but three proteins could be attributed to either one of the following protein classes: (1) association with cell membranes, (2) being part of the intracellular compartment of cells (organelles and cytoplasm), (3) attached to the cell surface (equivalent to extracellular matrix), or (4) serum or plasma, respectively. Proteins of the latter class were furthermore ascribed, whenever possible, to either one of the serum or blood plasma factor apolipoprotein, coagulation, complement, or immunoglobulin. The protein origin was, therefore, either cellular or serum/plasma, and a protein category was given directly by the class for classes 1–3, and by the serum/blood plasma factor for class 4.

The cluster of differentiation (CD) annotation was used as a surrogate for the cell origin of cEV for proteins of the cell membrane category. Annotations were decided according to the Human Cell Differentiation Molecules organization (hcdm.org), and the normalized mRNA expression levels in single cells from the human protein atlas (www.proteinatlas.org) accessed on 26 December 2021. Basement membrane-specific heparan sulfate proteoglycan core protein (HSPG2) was added as an endothelial cell-specific marker.

A special focus in our study was on platelets, as they are regarded as the most vulnerable cells to mechanical forces. The platelet proteome that was published by Burkhart et al. [25] was, therefore, used to annotate separately the possible origin of cEVs from platelets. An additional set of annotations were keratin (potential contaminant), blood microparticle, and exosome GO annotation. All annotations are provided in the Appendix A.

### 4.9. Differential Protein Abundance Testing

Label-free protein abundances were calculated from the sum of the intensities of the three most intense peptides of each protein group (Top3 approach), after summing two injections (mass spectrometry replicates) and normalizing the peptide intensities by variance stabilization (*vsn R* package). PFP and PPP plasma types were considered as different experimental sets and normalized independently for most of the study, except when comparisons were made between the two preparation methods, in which case all the samples were normalized together. The missing peptide intensities were imputed in the following manner: if at least two values were missing in one group of technical replicates, then these values were replaced by drawing random numbers from a Gaussian distribution of width 0.3 × sample standard deviation and centered at the sample distribution mean minus 2.5 × sample standard deviation; otherwise the missing value was replaced by the method of maximum likelihood estimation (MLE, *MSnbase R* package). The imputed Top3 protein intensities were called iTop3 in all accompanying documents. Differential abundance tests between each patient’s C and PTS samples of same plasma type were performed using empirical Bayes statistics (*limma R* package) on log-2 transformed iTop3 intensities, provided the protein groups were detected at least in one sample triplicate. Protein abundance differences were reported as the difference between the log2-transformed iTop3 intensities (log2fc), and adjusted *p*-values accounting for multiple testing were calculated using the FDR-controlled Benjamini and Hochberg correction (R base function *p.adjust*). Significance of the differential expression was defined by the combined criteria of |log2fc| ≥ 1 and adjusted *p*-value ≤ 0.05, such that the adjusted *p*-value must be zero for |log2fc| = 1 and 0.05 for asymptotically large fold changes. The curvature of the significance curve in between the extrema was determined by the overall variance. In order to overcome the stochasticity that was introduced by imputation, the imputation and significance test were repeated 20 times. Only those protein groups that were consistently reported as significantly differentially expressed throughout the imputation cycles were accepted as truly significant. The protein groups that were reported by MaxQuant as being identified only by site were excluded from statistical testing and downstream data evaluation. Contaminants of non-human origin were subsequently discarded. For comparisons between PPP and PFP, the post-hoc ANOVA tests were performed [30] with the Tukey’s honestly significant difference test using R base function TukeyHSD.

### 4.10. Functional Analysis of Proteins

For functional analyses, we collapsed the protein groups representing the same gene product (proteins annotated with same gene name in Uniprot) into one entry by summing their corresponding iTop3 intensities, provided there was at least one detection in the replicate group. The median of the log2 of the aggregated intensities of the three technical replicates was then used in the rest of the analysis and, where it could be determined, a corresponding median intensity value for each donor, mode of transport, and plasma type. Furthermore, an intensity that was representative of each protein subclass was calculated as the average of the log2 iTop3 intensities of the protein members. The same was done for the additional set of annotations.

Spearman’s rank correlations were calculated between a set of features including these representative intensities and transport metrics, blood count, and ZetaView^®^ measurements. The correlation coefficient rho was recorded only if the test *p*-value was ≤ 0.05, otherwise it was set to 0; the matrix of resulting correlation coefficients was used to perform an unsupervised clustering analysis.

Spearman rank correlations between the gene product intensities and the transport metrics TK, RMS, and VDV were then calculated separately for the PFP_C and PFP_PTS groups. A correlation coefficient rho was returned only if there were at least three gene product intensity values that were available in this group and considered significant if *p*-value ≤ 0.05. For each plasma type/transport group and transport metric, the gene products were ranked by 1 minus *p*-value, multiplied by the sign of the rho correlation, so that the proteins highly correlating with the transport metric were at the top of the list, and those most strongly anti-correlating at the bottom. For those lists where more than one transport metric was included, the lowest transport metric rank value was used for each gene product. The ranked lists were submitted to a statistical enrichment test of the gene ontology (GO) terms biological process and cellular component using the online PANTHER classification system [31], applying a 1% FDR control for multiple testing correction. The GO term lists were filtered by preferring those terms with the smallest *p*-values and highest number of significantly correlating gene products. Furthermore, only GO terms that were unique to one enrichment test were kept. To account for redundancy, we filtered the GO terms to the smallest possible list explaining all the involved gene products. With the resulting gene product lists, protein interaction network analyses were performed on the STRING database (string-db.org).

### 4.11. Defining EV-Associated Protein Transport Markers by Feature Selection Method

The least absolute shrinkage and selection operator (Lasso) algorithm was applied [32] in order to determine the selection of cEV proteins whose intensities discriminate the best between PTS and C. The calculations were performed in *R* using the *glmnet* package [33], both with an elastic net penalty of 0 (pure Lasso), in order to find a shortest selection, as well as with 0.5 (elastic net), in order to account to some extent for correlated variables. The optimal overall strength of the coefficient penalty, the parameter lambda, was determined by the “leave-one-out” cross validation method, whereby the model was calculated on *n-1* samples, and the miss-classification error on the remaining sample. The process is repeated *n* times, and the optimized lambda was determined from the minimum mean error. The response function aimed to take either the value of 0 for C, or 1 for PTS. The starting data set was, as for the functional analysis, all gene products with their corresponding median intensity per donor, and transport mode; here however, only the complete data (no missing intensities anywhere) were considered, so that a total of 349 gene products were retained for the analysis.

## 5. Conclusions

We can summarize that mechanical forces occurring during human carrier transport might affect the composition of cEV proteome profile by influencing plasma membrane reorganization and release of EV by an ectosomal pathway from blood cells. In contrast, PTS rather activated endosomal pathways due to the transport metric correlations with proteins of organelle origin. However, these processes might be biased by the cEV composition and stability of each person, which seems to be a consequence of health condition, other individual traits, and pre-analytical impacts during blood sample procurement.

## Figures and Tables

**Figure 1 ijms-23-04515-f001:**
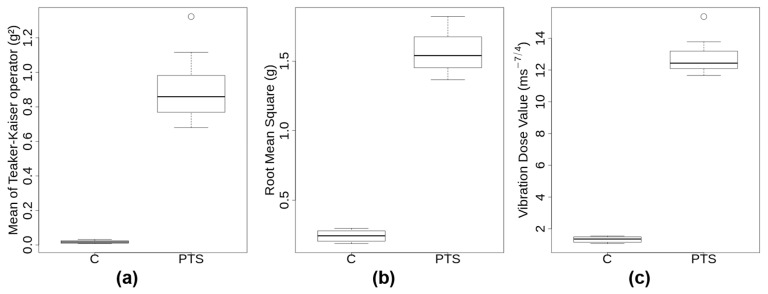
Boxplot representation of the transport metrics. The 12 C values were compared to the 12 PTS values for each transport metrics TK (**a**), RMS (**b**), and VDV (**c**) with C on the left and PTS on the right of each graph. The mean of Teaker–Kaiser operator for C ranged from 0.008 to 0.032 g^2^. All three metrics showed highly significant differences between C and PTS with Welch’s *t*-test *p*-values of 3.5 × 10^−9^ for mean of Teaker–Kaiser operator, 1.5 × 10^−13^ for root mean square, and 1.2 × 10^−13^ for vibration dose value, respectively.

**Figure 2 ijms-23-04515-f002:**
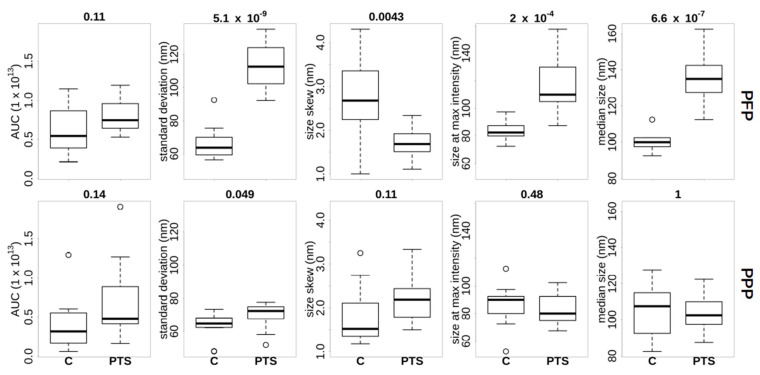
Extracted features from ZetaView^®^ nanoparticle size distributions for PFP (**upper row**) and PPP (**lower row**) samples. The samples are further separated by transport mode C (**left boxes**) or PTS (**right boxes**). From left to right the following features were extracted from the ZetaView^®^ size distribution: area under the curve (AUC), standard deviation of the particle size distribution, skewness of the size distribution, the size at maximum intensity, and the median particle size. The number above each boxplot is the *p*-value of the Welch’s *t*−test between the C and PTS values.

**Figure 3 ijms-23-04515-f003:**
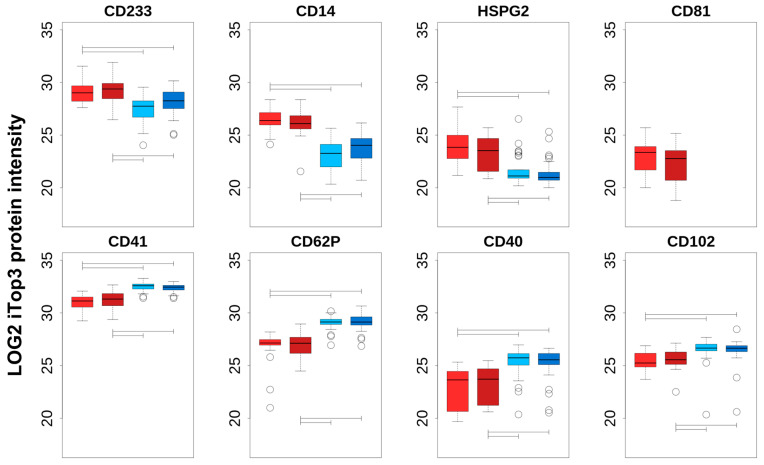
Abundance comparison of cell-type-specific proteins. Each boxplot shows, on the same scale, the spread of log2 iTop3 intensities of cell-type-specific proteins (indicated above the plot), for, from left to right, PFP_C (light red), PFP_PTS (dark red), PPP_C (light blue), and PPP_PTS (dark blue). Segments within the plots indicate when the one-way ANOVA post hoc pair-wise tests between the groups were significant (*p* ≤ 0.01). Protein intensities from all the samples have been normalized together. The top row of plots presents cells that were enriched in PFP with erythrocytes represented by *CD233*, macrophages by *CD14*, endothelial cells by *HSPG2*, and *CD81* being a more ubiquitous marker that is generally considered to represent exosomes. The cell-types that are represented in the second row of plots is platelets with *CD41* and *CD62P*, and lymphocytes and monocytes with *CD40* and *CD102*, respectively.

**Figure 4 ijms-23-04515-f004:**
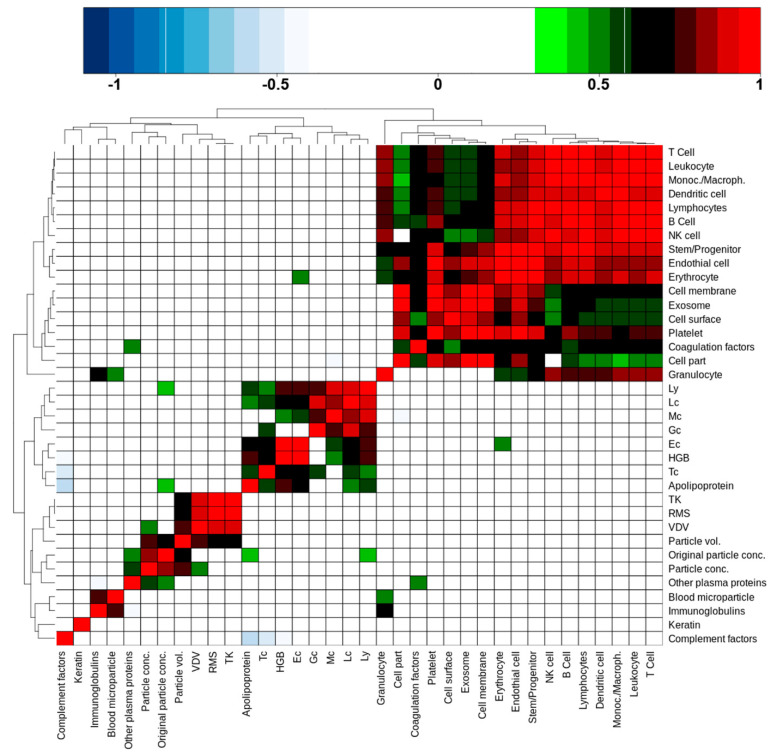
Unsupervised clustering of Spearman rank correlation rho values that were calculated from a variety of quantified features using PFP samples. The following features that were measured for each donor went into the correlation analysis: ZetaView^®^ nanoparticle tracking numbers (original particle concentration, non-corrected particle concentration and particle volume), blood cell counts of monocytes (Mc), lymphocytes (Ly), leukocytes (Lc), granulocytes (Gc), platelets (Tc), erythrocytes (Ec), hemoglobin concentration (HGB), transport metrics (VDV, RMS, TK), and label-free proteomics-determined protein category and cell type-specific marker intensities. The resulting correlation coefficients with *p*-value ≤ 0.05 are color-coded as shown on top of figure. All values are given in Appendix A.

**Figure 5 ijms-23-04515-f005:**
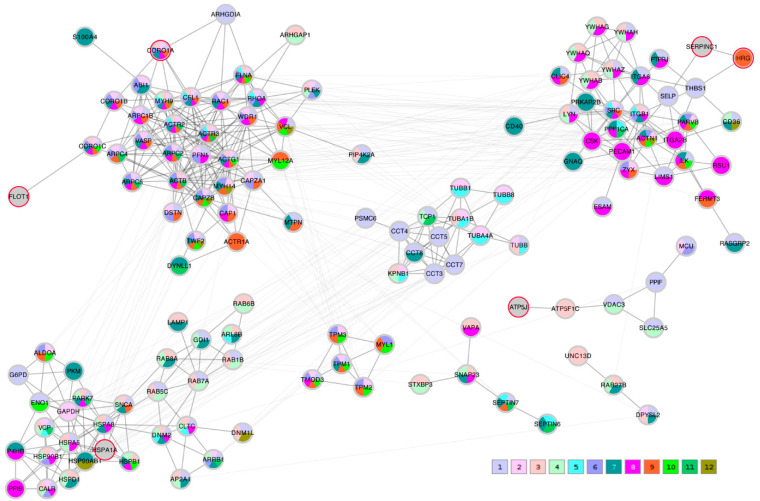
STRING protein-protein interaction network composed of proteins that were significantly correlating with all C transport metrics (C_TK/RMS/VDV) and the Lasso proteins. Edges represent a STRING combined score ≥ 0.7; they are drawn thick within community clusters, and thin across community clusters. The Lasso proteins are marked with a red border (gray otherwise). Node colors at the bottom right refer to GO terms given in Table 4, with 1–6 standing for biological process and 7–12 cellular component.

**Figure 6 ijms-23-04515-f006:**
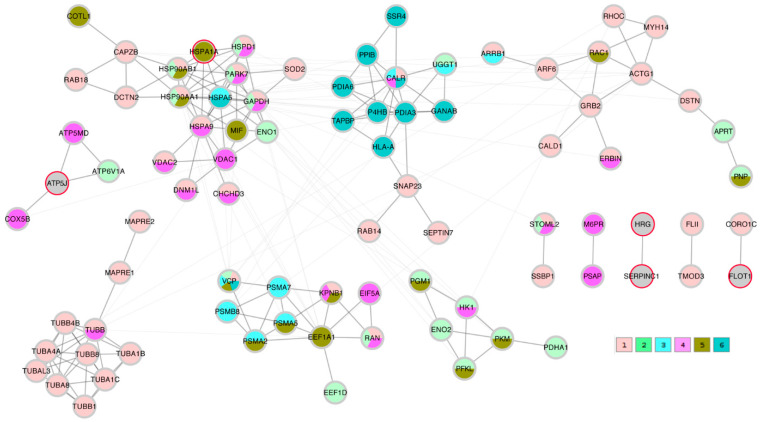
STRING protein-protein interaction network composed of proteins that were significantly correlating with the PTS metrics TK and RMS (PTS_TK/RMS) and the Lasso proteins. Edges represent a STRING combined score ≥ 0.7; they are drawn thick within community clusters, and thin across community clusters. Lasso proteins are marked with a red border (gray otherwise). Node colors at the bottom right refer to GO terms given in Table 4, with 1–3 standing for biological process and 4–6 cellular component.

**Figure 7 ijms-23-04515-f007:**
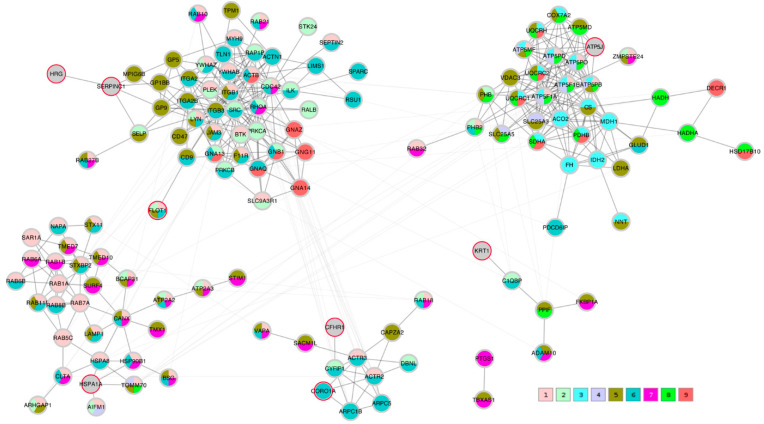
STRING protein-protein interaction network composed of proteins that were significantly correlating with the PTS transport metric VDV (PTS_VDV) and the Lasso proteins. Edges represent a STRING combined score ≥ 0.7; they are drawn thick within community clusters, and thin across community clusters. Lasso proteins are marked with a red border (gray otherwise). Node colors at the bottom right refer to GO terms given in Table 4, with 1–4 standing for biological process and 5–9 cellular component.

**Table 1 ijms-23-04515-t001:** Number of protein groups quantified in all cEV isolates.

**(a)** Classification by origin and category								
**Protein Origin**	**Protein Category**	**Combined Data**	**PFP**		**PPP**		**Plasma**	
			**C**	**PTS**	**C**	**PTS**	**unique-to-PFP**	**unique-to-PPP**
Cellular	Cell membrane	772	561	570	746	750	16	179
	Cell part	1030	723	752	991	992	23	248
	Cell surface	138	106	110	129	129	7	26
	Total cellular	1940	1390	1432	1866	1871	46	453
Serum/plasma	Apolipoprotein	19	19	19	18	19	0	0
	Coagulation factor	23	23	23	23	23	0	0
	Complement factor	26	26	26	26	26	0	0
	Immunoglobulin	75	73	74	69	71	4	1
	Other	58	56	56	56	56	2	2
	Total serum/plasma	201	197	198	192	195	6	3
Unknown		3	3	3	3	3	0	0
	Total	2144	1590	1633	2061	2069	52	456
**(b)** Cell type specific proteins (markers) *.								
**Cell Origin**		**Combined Data**	**PFP**		**PPP**		**Plasma**	
			**C**	**PTS**	**C**	**PTS**	**unique-to-PFP**	**unique-to-PPP**
T Cell		41	37	38	37	38	3	3
B Cell		36	32	33	32	33	3	3
NK cell		30	27	28	27	28	2	2
Dendritic cell		18	16	16	15	16	2	2
Monoc./Macroph.		50	44	45	44	46	4	5
Granulocyte		35	31	32	31	33	2	3
Platelet		34	33	33	34	34	0	1
Erythrocyte		13	13	13	12	12	1	0
Endothial cell		39	36	36	35	35	4	3
Stem/Progenitor		32	30	30	28	28	4	2
Total markers		70	63	64	62	64	6	6

* Most proteins have more than one cell type specificity.

**Table 2 ijms-23-04515-t002:** cEV-associated cell-type specific proteins that were used to assess cell-type specific damage by way of transport.

CD	GN	nTPM, Cell Type *	Functional Annotation Excerpt from uniport.org
CD14	*CD14*	Mp = 653 Mc = 285	Mediates the innate immune response to bacterial lipopolysaccharide (LPS)
CD40	*CD40*	Mc = 242 Bc = 148 Ec = 62	Transduces TRAF6- and MAP3K8-mediated signals that activate ERK in Mp and Bc, leading to induction of immunoglobulin secretion
CD41	*ITGA2B*	Pl Gc = 65 Ec = 2	Part of receptor for fibronectin, fibrinogen, plasminogen, prothrombin, thrombospondin and vitronectin
CD62P	*SELP*	Pl Ec = 52 Tc = 9 Gc = 8	Mediates the interaction of activated Ec or Pl with Lc
CD81	*CD81*	Ec = 345 Mp = 278 Dc = 267 Tc = 137 Bc = 88 Gc = 82	Structural component of specialized membrane microdomains known as tetraspanin-enriched microdomains, which act as platforms for receptor clustering and signaling.
CD102	*ICAM2*	Nk = 114 Ec = 109 Tc = 60 Mp = 57 Bc = 48 Mc = 26	Mediates adhesive interactions important for antigen-specific immune response, NK-cell mediated clearance, lymphocyte recirculation, and other cellular interactions important for immune response and surveillance
CD233	*SLC4A1*	Ery = 1623	Major integral membrane glycoprotein of the erythrocyte membrane
HSPG2	*HSPG2*	Ec = 323	Role in vascularization, basement membrane localization

* Normalized mRNA expression levels in single cells (nTPM) as published on the Human Protein Atlas organization website accessed on 26 December 2021 (https://www.proteinatlas.org/) in combination with information from the *Human Cell Differentiation Molecules* organization (hcdm.org). Bc = B-cells, Dc = dendritic cells, Ec = Endothelial/Epithelial cells, Ery = Erythrocytes, Gc = Granulocytes, Mc = Monocytes, Mp = Macrophages, Nk = natural killer cells, Pl = Platelets, Tc = T-cells.

**Table 3 ijms-23-04515-t003:** Correlations of protein group intensities with transport metrics. The numbers of protein groups significantly correlating (*p* ≤ 0.05) with transport metrics are reported on the left for positive correlation, and on the right for anti-correlation. C and PTS were both pooled (C + PTS) and considered separately.

	Positive Correlations			Negative Correlations		
Transport Metrics	C + PTS	C	PTS	C + PTS	C	PTS
TK only	14	43	31	1	3	2
RMS only	1	3	15	0	2	3
VDV only	10	7	158	2	1	20
TK + RMS	12	12	144	5	4	16
RMS + VDV	0	6	0	1	2	0
TK + VDV	8	36	32	0	4	0
TK + RMS + VDV	27	181	4	2	11	0
Total TK	61	202	194	8	19	20
Total RMS	40	202	163	8	19	19
Total VDV	45	230	194	5	18	20

**Table 4 ijms-23-04515-t004:** Significantly enriched GO terms in correlating transport metric lists. The network color code refers to Figure 5, Figure 6 and Figure 7.

Network Color	Biological Process	*p*-Value	# of Genes	Network Color	Cellular Component	*p*-Value	# of Genes
C_TK/RMS/VDV							
1	regulation of cellular component organization	1.3 × 10^−6^	82	7	plasma membrane bounded cell projection	9.8 × 10^−7^	63
2	cytoskeleton organization	3.8 × 10^−7^	52	8	anchoring junction	2.1 × 10^−10^	59
3	establishment of localization in cell	3.7 × 10^−8^	51	9	actin cytoskeleton	1.2 × 10^−9^	44
4	nitrogen compound transport	1.2 × 10^−4^	40	10	myofibril	2.6 × 10^−7^	19
5	cell cycle	9.5 × 10^−5^	23	11	chromosome	1.1 × 10^−4^	15
6	actin filament-based process	2.2 × 10^−7^	43	12	cluster of actin-based cell projections	1.9 × 10^−4^	12
**PTS_TK/RMS**							
1	organelle organization	5.0 × 10^−5^	56	4	organelle envelope	6.7 × 10^−5^	31
2	nucleobase-containing compound metabolic process	6.9 × 10^−5^	22	5	ficolin-1-rich granule	1.2 × 10^−4^	16
3	proteasomal protein catabolic process	3.5 × 10^−5^	9	6	endoplasmic reticulum protein-containing complex	1.0 × 10^−4^	11
**PTS_VDV**							
1	cellular localization	2.8 × 10^−5^	63	5	intrinsic component of membrane	8.3 × 10^−6^	76
2	intracellular signal transduction	4.8 × 10^−5^	37	6	cell junction	7.9 × 10^−5^	64
3	cellular respiration	2.2 × 10^−6^	19	7	organelle sub compartment	2.5 × 10^−4^	35
4	mitochondrial transmembrane transport	5.6 × 10^−5^	12	8	mitochondrial protein-containing complex	3.9 × 10^−8^	21
				9	catalytic complex	2.9 × 10^−4^	19

## Data Availability

The proteomics data (consisting of 288 LC-MS/MS files) have been deposited to the ProteomeXchange Consortium via the PRIDE [35] partner repository with the dataset identifier PXD033117.

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
