# Peer review of "Effect of Sample Transportation on the Proteome of Human Circulating Blood Extracellular Vesicles"

_ijms, 2022, doi:10.3390/ijms23094515_

Round 1

Reviewer 1 Report

In the article: “Effect of Sample Transportation on the Proteome of Human Circulating Blood Extracellular Vesicles”  the authors explained the importance of circulating extracellular vesicles (cEV) in several biological process and their potential application as therapeutical agents”.   

The authors well explain the rational of the study, displayed the performed experiments and clearly discussed them. However, we would like to invite the authors  to clarify some minor points:

  1. Line 60: the authors say: “ moreover, characteristic profiles of cEV were established, allowing the determination of the cell origin profile and confirming that the method of freezing of blood samples has a relevant impact on cEV proteome integrity”, please try to better explain this concept.
  2. Line 43: the authors explain that EVs play a role in several biological process, such as inflammation and coagulation. They should could also hint to recent hypotheses, such as a role in the differentiation of mesenchymal cells such as displayed in this paper: Vassallo V, Tsianaka A, Alessio N, Grübel J, Cammarota M, Tovar GEM, Southan A, Schiraldi C. Evaluation of novel biomaterials for cartilage regeneration based on gelatin methacryloyl interpenetrated with extractive chondroitin sulfate or unsulfated biotechnological chondroitin. J Biomed Mater Res A. 2022 Jan 28. doi: 10.1002/jbm.a.37364. Epub ahead of print. PMID: 35088923.
  3. Line 80: the authors say: “They reported an increase in procoagulant activity, along with an increase of annexin-V positive vesicles”. There are other biomarkers useful to this aim? Besides proteomics is there any other in vitro assay that can be performed? The authors should better clarify this point.
  4. Line 575: please the authors could specify the age and the sex of human donors.
  5. Figure 1A: Standard Deviation is not clear visible in the graph. Please try to improve the figure.
  6. Tables: check if the font is the same for all the lines.

Reviewer 2 Report

In their paper entitled “Effect of Sample Transportation on the Proteome of Human Circulating Blood Extracellular Vesicles”, the Authors report the results of an analysis aimed at investigating modifications of the composition and size distribution of extracellular vesicles (EVs) during their transport from the wards to the laboratories, by either human carriers (C), or pneumatic tube systems (PTS). They found and report a correlation between proteomic protein intensity and kind of transport: in particular, transport by human carriers seems to influence release of EVs through the ectosomal pathway, while transport by PTS seems to stimulate the endosomal pathway. As a whole, their results suggest that human carriers are preferable. Moreover, the Authors report that platelet-free plasma (PFP) was found to be more advisable as the starting material for any further analysis.

The paper is certainly of interest, since EVs are increasingly considered as useful biomarkers in pathology and they are very often transferred from clinical wards to the bench; however, as the Authors underline, the effects of transporting these biological samples is not yet completely clear. In addition, from the results of the paper, it seems that variability of EV composition also depends on the source of the individual blood sample (i.e. it is somehow donor-specific).

Methods and Results are clearly described. However, a further point that the Reader might like to see discussed is the fact that, if cEVs are sensitive to acceleration due to transport, probably one is allowed to think that, during the purification steps based on high speed centrifugation, particles might be significantly modified both for size and composition.

Author Response

This manuscript is a resubmission of an earlier submission. The following is a list of the peer review reports and author responses from that submission.

Round 1

Reviewer 1 Report

Ref.30 page 20 does not contain the description of "Tukey’s Honestly Significant Difference test" used to compare PPP and PFP, please change it.

Author Response

Response to Referee 1   Thank you for spotting the issue with Ref 30. This reference does not indeed describe Tukey’s Honestly Significant Difference test directly; rather, it is a reference to a publication where ANOVA tests have been performed in a similar manner by our laboratory. We have therefore corrected lines 702-703 in the following manner:   For comparisons between PPP and PFP, the post-hoc ANOVA tests were performed [30] with the Tukey’s Honestly Significant Difference test using R base function TukeyHSD.

Reviewer 2 Report

The study of sample transportation influence to proteome of EVs is interesting and definitely deserves attention. However, due to serious shortcomings, I have to give the article a negative review.

 In particular, the used method for isolation of EVs is highly questionable. After centrifugation of blood at 1500 g for 10 min (speed is very high for first centrifugation!), authors centrifuged plasma at 16000g for 40 min and believe that the sediment contains vesicles. However, to precipitate vesicles requires a speed of 100,000 g at 1.5h!

To estimate particle count and size by NTA authors diluted plasma and believe that the registered signals obtained from vesicles.

As result, in manuscript is absent evidence of isolated vesicles. for expamle, TEM, Cryo-Electron Microscopy, row data of NTA, etc

After such illiterate manipulations, it makes no sense to discuss the results of MS.

Author Response

Response to Reviewer 2 Comments

Point 1: The study of sample transportation influence to proteome of EVs is interesting and definitely deserves attention. However, due to serious shortcomings, I have to give the article a negative review.

Response 1: We are happy to read that Reviewer 2 agrees that our study on the influence of transportation on vesicles is interesting and deserves attention. We discuss the “shortcomings” under the next points.

Point 2: In particular, the used method for isolation of EVs is highly questionable. After centrifugation of blood at 1500 g for 10 min (speed is very high for first centrifugation!), authors centrifuged plasma at 16000g for 40 min and believe that the sediment contains vesicles. However, to precipitate vesicles requires a speed of 100,000 g at 1.5h!

Response 2: We thank reviewer 2 for the comment and respectfully would like to draw his/her attention to the method section and the accompanying supplementary materials. We disargree with Reviewer 2's own suggestion on how to precipitate vesicles as this would yield unsuitable material for our context, as explained below.

  1. i) A first centrifugation of blood with accelerations (it’s not speed!) of more than 1000 g is actually standard practice in order to sediment blood cells including platelets, resulting in platelet-poor, but not platelet-free plasma (PPP). We used PPP to isolate vesicles in order to see if transportation had an effect on the formation of cell debris, which would still be present in PPP (control experiment). For the most important conclusions of our work, we did a second centrifugation at 16’000 g for 2 minutes in order to sediment cell debris, remaining platelets and larger vesicles, which produces a supernatant commonly refered to as platelet-free plasma (PFP). This kind of procedure is standard practice to prepare PFP and tested in many laboratories using different assays [Ref. 11 in manuscript; Jy et al. (2004) Thromb. Haemost 2, 1842-1851; Trummer et al. (2009) Blood. Coagul. Fibrinolysis 20, 52-56; Issman wt al. (2013) PLoS One 8, e83680; Yuana et al. (2015) J. Extracell. Vesicles 4, 29260]. The 40-minute centrifugation at 16’000 g was then used to pellet vesicles. We have modelled the sedimentation behavior of vesicles and platelets under these conditions in an earlier publication [Ref. 11 in manuscript]. We determined there that with this protocol we lose part of the smallest vesicles having diameters <200 nm and removed larger ones with diameter >500 nm. We confirmed the modelling with cryo-TEM pictures in the same publication.

  1. ii) The suggested protocol by Reviewer 2 refers to the isolation of the smallest vesicles with diameter <200 nm. We tried such protocols on PFP with the result that the pellet consisted essentially only of albumin, immunoglobulins and apolipoproteins, hence can’t be applied to a study of blood plasma vesicle proteomes.

In this study, we discuss and show results (supplementary materials) using an alternative method for the isolation of all vesicle size classes, which is based on size-exclusion chromatography (SEC), and compared it with our centrifugation protocol. We show with quantitative proteomics that our centrifugation protocol performs actually much better than SEC in the purification of vesicles (as represented by cellular proteins such as cluster of differentiation markers) from contaminating plasma proteins, and has a better reproducibility than SEC. Last but not least, we present the quantification of 740 cell membrane proteins found in at least 2 PFP samples; yet cell membrane proteins are hugely under-represented in shotgun proteome studies of whole cell lysates. Furthermore, in reference 11 of our manuscript we had demonstrated that nuclear and ribosomal proteins are clearly under-represented in our vesicle preparations compared to whole cell lysates. All together, these facts demonstrate convincingly that with our centrifugation protocol we do indeed isolate extracellular vesicles, and not cells.

Point 3: To estimate particle count and size by NTA authors diluted plasma and believe that the registered signals obtained from vesicles.

As result, in manuscript is absent evidence of isolated vesicles. for expamle, TEM, Cryo-Electron Microscopy, row data of NTA, etc.

Response 3: First, we used a commonly applied NTA protocol for vesicle size distribution studies in plasma [for instance Khan et al. (2021) Brain Behav Immun. 2021;92:165-183 pops-up ahead of 81 articles with a pubmed search using ((NTA) AND (vesicles)) AND (plasma)]. Again, the used EV isolation method was developed and published earlier by us [Ref. 11 in manuscript], where we showed cryo-TEM pictures. In this study, we applied the exact same vesicle isolation protocol like in Ref. 11, it was carried out by the same operator using the exact same tubes and equipment, and the list of quantified proteins overlaps largely with the one described in Ref. 11. In our believe, this is enough evidence to assume that we look at the same thing here like published earlier.

We do not understand what Reviewer 2 means with row data of NTA. If he/she refers to “raw data” then we claim that this is sufficiently illustrated in figure 2 of the main manuscript, figure S2 andTable S5 in the supplemtary material. Of course, we can provide the Excel sheet with all data points, although we haven’t seen any NTA data containing publication where such raw data was provided.

Point 4: After such illiterate manipulations, it makes no sense to discuss the results of MS.

Response 4: Reviewer 2 dismisses the premises of our study, we cannot help noticing, without providing substantiated and literate criticism. We are therefore not able improve anything concrete, as requested, at this stage. We appreciate Reviewer 2’s comments. However, for all the above mentioned reasons, we respectfully disagree with them.

Round 2

Reviewer 2 Report

The authors' arguments on the quality of the obtained preparations of extracellular vesicles are not convincing. The body of the manuscript lacks data from cryo-EM, NTA, etc.
Please remove me from the list of reviewers